# Shotgun Proteomics of Isolated Urinary Extracellular Vesicles for Investigating Respiratory Impedance in Healthy Preschoolers

**DOI:** 10.3390/molecules26051258

**Published:** 2021-02-26

**Authors:** Giuliana Ferrante, Rossana Rossi, Giovanna Cilluffo, Dario Di Silvestre, Andrea Brambilla, Antonella De Palma, Chiara Villa, Velia Malizia, Rosalia Gagliardo, Yvan Torrente, Giovanni Corsello, Giovanni Viegi, Pierluigi Mauri, Stefania La Grutta

**Affiliations:** 1Department of Health Promotion, Mother and Child Care, Internal Medicine and Medical Specialties, University of Palermo, Piazza delle Cliniche, 2, 90127 Palermo, Italy; giuliana.ferrante@unipa.it (G.F.); giovanni.corsello@unipa.it (G.C.); 2National Research Council of Italy, Proteomics and Metabolomics Unit, Institute for Biomedical Technologies, ITB-CNR, Via Fratelli Cervi, 93, Segrate, 20090 Milan, Italy; rossana.rossi@itb.cnr.it (R.R.); dario.disilvestre@itb.cnr.it (D.D.S.); antonella.depalma@itb.cnr.it (A.D.P.); 3National Research Council of Italy, Institute for Biomedical Research and Innovation (IRIB), Via Ugo La Malfa 153, 90146 Palermo, Italy; giovanna.cilluffo@irib.cnr.it (G.C.); velia.malizia@irib.cnr.it (V.M.); rosalia.paola.gagliardo@irib.cnr.it (R.G.); giovanni.viegi@irib.cnr.it (G.V.); stefania.lagrutta@irib.cnr.it (S.L.G.); 4Stem Cell Laboratory, Department of Pathophysiology and Transplantation, University of Milan, Unit of Neurology, Fondazione IRCCS Ca’ Granda Ospedale Maggiore Policlinico, Centro Dino Ferrari, 20122 Milan, Italy; andreabrambilla1991@gmail.com (A.B.); chiara.villa2@unimi.it (C.V.); yvan.torrente@unimi.it (Y.T.); 5Clinical Proteomics Laboratory c/o ITB-CNR, CNR.Biomics Infrastructure, Elixir, Via Fratelli Cervi, 93, Segrate, 20090 Milan, Italy

**Keywords:** extracellular vesicle, urine fractionation, proteomics, forced oscillation technique, preschooler healthy children

## Abstract

Urine proteomic applications in children suggested their potential in discriminating between healthy subjects from those with respiratory diseases. The aim of the current study was to combine protein fractionation, by urinary extracellular vesicle isolation, and proteomics analysis in order to establish whether different patterns of respiratory impedance in healthy preschoolers can be characterized from a protein fingerprint. Twenty-one 3–5-yr-old healthy children, representative of 66 recruited subjects, were selected: 12 late preterm (LP) and 9 full-term (T) born. Children underwent measurement of respiratory impedance through Forced Oscillation Technique (FOT) and no significant differences between LP and T were found. Unbiased clustering, based on proteomic signatures, stratified three groups of children (A, B, C) with significantly different patterns of respiratory impedance, which was slightly worse in group A than in groups B and C. Six proteins (Tripeptidyl peptidase I (TPP1), Cubilin (CUBN), SerpinA4, SerpinF1, Thy-1 membrane glycoprotein (THY1) and Angiopoietin-related protein 2 (ANGPTL2)) were identified in order to type the membership of subjects to the three groups. The differential levels of the six proteins in groups A, B and C suggest that proteomic-based profiles of urinary fractionated exosomes could represent a link between respiratory impedance and underlying biological profiles in healthy preschool children.

## 1. Introduction

Physiological changes throughout childhood characterize lung function, which is at least in part influenced by perinatal factors, including prematurity. Significantly, Late Preterm (LP, 34–36 weeks’ gestational age, GA) children without clinical lung disease may show deficits in lung function that may persist throughout infancy [1]. Indeed, increased respiratory impedance was reported in healthy children aged 3–7 years born LP in comparison with age-matched healthy term-born children [2]. Studying the underlying biological profiles may provide additional knowledge on this functional picture. In this context, omics technologies have shown the potential to discriminate healthy schoolchildren from those with respiratory diseases, unlike lung function tests such as spirometry [3]. No data have been published so far with regard to healthy preschool children and their respiratory impedance.

Concerning lung investigations, advances in proteomics have led to the discovery of several new protein markers, allowing the characterization of proteins from both lung tissue and urine which are involved in respiratory diseases [4,5,6]. In particular, the fractionation of urine samples allows the isolation of extracellular vesicles, and the characterization of their protein profiles is useful to stratify asthmatic patients by specific biomarkers, such as galectin-3 binding protein [6]. Urine is becoming one of the most attractive bio-fluids in clinical proteomics because of its high protein and peptide content and also because procuring it is easy and non-invasive [7]. Indeed, a specific fraction of urine represented by extracellular microvesicles (EVs) has attracted increasing research interest, given the high presence in them of proteins involved in intercellular communication and molecular pathways [8]. Previous studies demonstrated that the proteomic profiling of urine samples might be applicable to pediatric age. In particular, urine proteomic applications in children suggested their potential in discriminating between healthy subjects and those with obstructive sleep apnea [9]. Of note, the proteomic profiles of urine in healthy preschool children have not been investigated so far. 

The aim of the present study was to verify that proteomics analysis allows protein profiles useful to characterize the eventual different patterns in healthy preschool children born late preterm (LP) and full-term (T) and their respiratory impedance. To improve the specificity of our analysis, we combined proteomics analysis with the isolation of a specific protein urinary compartment, such as extracellular vesicles, in order to establish whether different patterns of respiratory impedance can be characterized from a proteomics fingerprint.

## 2. Results

### 2.1. Characteristics of Study Participants

A sub-sample of 21 randomly selected subjects (LP = 12; T = 9) for proteomic analysis was comparable to the entire sample of 66 children, since no difference was detected for personal characteristics and Forced Oscillation Technique (FOT) parameters (Appendix A). Among the 21 subjects, no significant differences were found between LP and T, except for height *Z*-score and birthweight, which were significantly lower in LP (Table 1). In the same way, in the total sample (*n* = 66), no differences were found for the *Z*-scores of Resistance of the respiratory system (Rrs) and Reactance of the respiratory system (Xrs) at 6, 8, and 10 Hz and area under the reactance curve (AX) between late preterm (LP) and term (T) either (data not shown).

### 2.2. Isolation of Urinary Extracellular Vesicles and Proteomics Analysis

Urinary proteins were fractionated by extracellular vesicles isolation using ultracentrifugation, and size assessed by dynamic light scattering, displaying a size distribution (Figure 1) with a peak occurring roughly at 150 nm typical for extracellular vesicles [10,11]. Urine yielded an amount of extracellular vesicles around 2–3 × 10^9^/mL (Figure 1). 

After protein extraction and tryptic digestion, urinary extracellular vesicles were analyzed by shotgun proteomics, based on nanoLC-MS/MS, and 1127 proteins were identified (Appendix A) in a wide range for both molecular weight and isoelectric point (Appendix A). Interestingly, each technical replicate showed both linear correlation and slope close to a theoretical value of 1 (Appendix A). From protein lists, an *n*x*m* matrix was prepared, consisting of identified proteins presenting a frequency >25% (out of 42 runs): hierarchical clustering placed children in three distinct groups (A, B and C) (Figure 2), while linear discriminant analysis (LDA) extracted 360 proteins (F ratio > 5 and *p*-value < 0.01) as discriminants (Appendix A). About 74 and 26% proved to be shared or related to a specific group, respectively (Appendix A). Identified proteins were compared to the Vesiclepedia database (http://microvesicles.org/ (accessed on 17 December 2020)), and more than 90% were known as EV proteins; in addition, gene ontology analysis confirmed the localization of identified proteins mainly as exosome and extracellular-like (Appendix A).

Identified proteins from A/B/C groups were compared, and differentially expressed proteins (DEPs) were extracted using a label free approach based on SpCs (Spectral Counts) value. In fact, SpC represents the total number of MS/MS spectra assigned to each protein and, consequently, it reflects protein relative abundance in each analyzed sample. In particular, the proteins differentially expressed in the three groups were semiquantitatively evaluated by the DAVE (differential average) and DCI (differential confidence index) algorithms from the MAProMa software. In this way, 232 distinct DEPs were identified: about 90% of them (212 proteins) were common to discriminants obtained from LDA (Appendix A). In addition, 15 of these proteins proved to be highly confident, in terms of statistical and differential analysis, for each group, and matched with discriminant by LDA; in particular, six proteins (Tripeptidyl peptidase I (TPP1), Cubilin (CUBN), SerpinA4, SerpinF1, Thy-1 membrane glycoprotein (THY1) and Angiopoietin-related protein 2 (ANGPTL2)) were found to be differently distributed among the three groups of children (Figure 3a). Additionally, the α-value algorithm made it possible to estimate the relative abundance of selected proteins (TPP1, Serpins and THY1) for each of the 21 subjects, allowing assignment to group A, B or C (Figure 3b and Appendix A) by a set of selected proteins. In particular, TTP1 and Serpin are higher in groups A and C, respectively, while in these groups THY1 is very low; on the contrary, group C is characterized by a similar level for the three proteins. Typing confirmation may be obtained by the other three proteins: CUBN distinguishes group A from B and C; and SerpinA4 and ANGPTL2 differentiate group C from A and B.

Furthermore, eleven CD (Cluster of Differentiation) markers were identified: among them, six showed confident statistical and differential values. Specifically, CDs always proved lower and higher in groups C and B, respectively (Figure 4).

Finally, in order to identify protein pathways in the three groups (A, B and C), selected descriptors of the three groups were plotted into the *Homo sapiens* protein–protein interaction network (Appendix A and Appendix A); in this way, it is possible to evaluate, in a simplified way, the differences of pathway expression between the groups [12]. Specifically, the modules related to the immune system, extracellular matrix (ECM), collagens and kinase proved to be mainly up-regulated in group B and downregulated in group C, while keratin, lipid metabolic process and carboxylic acid metabolic process proved to be up-regulated in group C (Figure 5).

### 2.3. Linking FOT Parameters and Proteomics

The *Z*-scores of resistance and reactance at 6, 8, and 10 Hz and AX in the three groups (A, B and C) are depicted in Figure 6. After multiple comparisons, groups A and C were statistically different for all FOT parameters, in particular, the *Z*-scores for resistance at Rrs6, Rrs8, and Rrs10 Hz and AX values in group A were higher than in group C. The *Z*-scores for reactance at Xrs6, Xrs8, and Xrs10 Hz values in group A were lower than in group C; groups A and B were statistically different for *Z*-scores scores reactance at Xrs8 and Xrs10 Hz values, which were lower in group A than in group B (Figure 6). 

## 3. Discussion

Here, we report the application of an innovative approach based on urinary proteomics to characterize the respiratory impedance in healthy preschoolers. Analyzing urine proteome, we were able to link the underlying biological profiles and different patterns of respiratory impedance observed in healthy preschoolers. Of note, the three proteome-based groups evidenced significantly different resistance, reactance and AX values measured by FOT, regardless of being LP or T. Specifically, we identified groups of children with different respiratory impedance based on the molecular stratification of fractionated urinary proteome. Unlike the entire urine sample used in previous studies by Becker et al. and Starodubtseva et al. [9,13], we utilized the extracellular vesicle fraction, which is considered the carrier of functional signals [14], making it possible to identify over 1100 proteins. Indeed, correct fractionation is confirmed by the level of typical free urinary proteins: we found that albumin and serotransferrin levels, free proteins in urine samples, were decreased 5–6 times with respect to the levels usually reported in the entire urine [13]. By contrast, typical exosomal proteins were detected at a higher level in urinary extracellular vesicle samples; for example, inter-alpha-trypsin inhibitors, such as ITIH4, increased 5–6 times compared to the level found in the entire urine [13]. Additionally, we identified 9 out of the top 10 proteins that are often detected in extracellular vesicles (exocarta website; http://exocarta.org/exosome_markers_new (accessed on 17 December 2020)). 

Overall, in our study, the biomarkers characterized in urinary extracellular vesicles corresponded to about 50% of urinary biomarkers identified by Becker et al. in children with obstructive sleep apnea [9]. In addition, six out of our 15 highly confident proteins were confirmed, including Cubilin (CUBN) and Angiopoietin-related protein 2 (ANGPTL2), which we proposed for typing proteomics groups (A, B and C) by an adapted alpha-value algorithm. Our proteomic data are in agreement with those published by Starodubseva and coworkers [13,15], as almost all their identified proteins in newborns correspond to those characterized in our study. The correspondence is quite high (about 70%) considering the so-called “core proteome of urine”, including 104 proteins, and it increases to 100% considering only proteins distinguishing newborns with and without respiratory infectious disorders [16].

By means of clustering analysis of microvesicle proteome profiles, three subgroups (named A, B and C) were identified in the overall sample of children. More importantly, the three proteome-based sub-groups had significantly different resistance, reactance and AX values measured by FOT. These findings indicate that different patterns of respiratory impedance were found, being slightly worse in group A than in groups B and C. Therefore, the analysis of fractionated proteome appeared to be informative in depicting the biological profiles underlying different patterns of respiratory impedance in healthy preschoolers.

To simplify the monitoring, we selected six proteins (Tripeptidyl peptidase I (TPP1), CUBN, SerpinF1, SerpinA4, Thy-1 membrane glycoprotein (THY1) and ANGPTL2) for typing the membership of subjects to the three groups (A, B and C), and three of them (TPP1, THY1 and SerpinA4) were applied in the alpha-value algorithm. TPP1, which proved to be higher in group A, is a peptidase localized in type II alveolar epithelial cells [17]. Interestingly, Ohlmeier et al. found that TPP1 showed high levels in mild to moderate chronic obstructive pulmonary disease [18]. CUBN, a plasma membrane receptor expressed on alveolar type II cells involved in endocytic update of vitamin D, proved to be higher in group A. The finding that the aforementioned urinary proteins proved to be higher in children with the worst FOT parameters (group A) might suggest their role as putative biomarkers of small airway impairment.

The two selected SerpinF1 (Pigment epithelium-derived factor, PEDF) and SerpinA4 (Kallistatin) were found to be higher in group C than in subjects in groups A and B. The role of SerpinF1 has been clarified in animal models, showing that this protein is able to inhibit eosinophilic airway inflammation, airway hyperresponsiveness and airway remodeling [19]. Similarly, SerpinA4 showed anti-inflammatory activity in animal models [20]. These findings in children with the best FOT parameters may be suggestive for a putative role of these two proteins in preserving lung function. 

Children in group B were characterized by higher levels of Thy-1 membrane glycoprotein (THY1), also known as CD90, and lower levels of angiopoietin-related protein 2 (ANGPTL2) than those in groups A and C. The role of THY1 is as a glycophosphatidylinositol anchored cell surface glycoprotein [21,22], and it has been recently characterized to be a key regulator of the WNT pathway, attenuating interstitial pulmonary fibrosis and promoting lung fibroblast apoptosis [23]. ANGPTL2 derived from lung epithelial cells has a protective role against fibrosis in lungs [24]. Then, we can conclude that the differential levels of the six proteins in groups A, B and C provide a characterization of the biological profiles underlying different patterns of respiratory impedance.

The application of network analysis allowed us to find a number of modules that showed a similar trend for groups A and B: specifically, modules related to cell adhesion, collagen, kinases, extracellular matrix (ECM) and the immune system showed the same level in groups A and B, and a higher one in group C. By contrast, only proteases and histones showed higher levels in group A. It is possible to speculate that the activation of these pathways, specifically ECM, the immune system and proteases, is related to inflammation pathways in the lung. Moreover, the detection of higher levels of endopeptidase inhibitors, complement and coagulation cascades, keratin and actin pathways in group C might explain the finding of the best FOT parameters in this group. Altogether, our results open new avenues for understanding the role of biological profiling underlying the respiratory impedance in healthy children. The lack of external validation and the small sample size are the main limitations of the current study. Therefore, our results must be considered preliminary and further studies are needed to confirm them in larger groups of children. It is to be pointed out that this is the first study involving healthy preschoolers integrating measurements of lung function and proteomics; for the latter, in addition, in view of the abundance of information retrievable from each subject, the sample size could be considered adequate for a proof of principle study. An important strength is the application of unbiased proteomics-based clustering, which identified three groups characterized by different patterns of respiratory impedance. The ongoing study will expand the internal validity of the current findings. Although the usual application of EVs is for investigating kidney-related diseases, recent studies evidenced that EVs can be used to investigate unrelated urogenital diseases, such as thyroid, bladder and pancreatic cancers [14,25], asthma [6] and neurological diseases [26].

## 4. Materials and Methods

### 4.1. Participants and Study Design

The current cross-sectional study is a part of the ongoing case-control longitudinal observational study “PREmaturely born preschool children-Asthma and Allergic Rhinitis” (PRE-AR). A total of 21 3–5-yr-old consecutively enrolled healthy children were preliminary evaluated: 9 born full-term (T) and 12 born LP. The local ethics committee approved the study (AOUP Paolo Giaccone, Palermo, Italy, *n*.9/2014), and written informed consent was obtained from parents or legal guardians prior to testing. This study was performed in accordance with the Declaration of Helsinki and Good Clinical Consent for publication. The approved study was entered into the central registration system ClinicalTrials.gov (identifier: NCT02636933). Inclusion criterion for both T and LP was 5 years old ≤ age ≥ 3 years old. Exclusion criteria for both T and LP were the following: (i) bronchopulmonary dysplasia (BPD); (ii) malformation of upper respiratory tract; (iii) immunological and/or metabolic and/or neurological diseases; (iv) any reported respiratory tract infection within 4 weeks prior to enrolment; (v) topical or systemic therapies with antibiotics, antihistamines and corticosteroids in the 30 days prior to enrolment; (vi) patients not able to perform lung function tests. 

Information on socio-demographic characteristics, maternal disease in pregnancy (urinary infections, gestational diabetes, pre-eclampsia), mode of delivery, neonatal respiratory support, bronchiolitis within the 1st year of life, history of upper respiratory infections and pneumonia ever and current environmental exposure proximity to a high traffic road <200 m/parental smoke/pet/mold) was obtained from parents through an interviewer-administered questionnaire. 

### 4.2. Forced Oscillation Technique (FOT)

Each subject underwent FOT measurement through a commercial device (Quark i2m^®^ Forced Oscillation Measurement system, Cosmed, Italy) based on a pseudo-random noise signal between 4 and 48 Hz. Measurements were performed according to international guidelines [27]. Only measurements with a 95% minimal coherence function were considered valid. Three acceptable measurements were taken, and the mean value was reported for each child at the frequencies of 6, 8, and 10 Hz. The obtained values were transformed into *Z*-scores according to reference equations [28].

### 4.3. Isolation of Urinary Extracellular Vesicle

About 100 mL of morning urine was collected and centrifuged at 17,000× *g* for 10 min at 4 °C; then, 5 mL of supernatant fractions was collected and subjected to ultracentrifugation at 200,000× *g* for 1 h at 4 °C to obtain extracellular vesicles for nanosight and proteomic analyses. 

### 4.4. Nanosight Extracellular Vesicle Analysis 

Isolation of EVs from urine samples was performed with a series of ultracentrifugations as already indicated above, replacing the resuspension buffer with 500 µL of phosphate-buffered saline PBS 1X filtered 0.1 µm. Evaluation of the vesicle size distribution profiling was performed with the Nanoparticle Tracking Analysis (NTA) technique (Nanosight NS300, Malvern Instruments Limited, Worcestershire, UK). The NTA measurement settings were set as follows: temperature 23.75 ± 0.5 °C; viscosity 0.91 ± 0.03 cP; measurement time of 60 s for each of the five repeated records per sample; infusion flow speed 30 and camera level 13. All extracellular vesicle samples were diluted with PBS 1X filtered 0.1 µm; the dilution factor was adapted in accordance with the initial concentration of the sample in order to perform all the analysis in a range of 20 to 120 particles per frame.

### 4.5. Proteomics Analysis of Urinary Extracellular Vesicle

The isolated EVs from urine samples were resuspended in 0.1 M ammonium bicarbonate (Sigma-Aldrich Inc., St.Louis, MO, USA company, city, state abbrev if USA, country), pH 7.9, and were trypsinized according to procedures previously reported [29], using 50 ± 0.5 µg of proteins from each sample.

One microliter (about 1 µg injected) of trypsin-digested mixtures was analyzed by nano-cromatography equipped with a cHiPLC-nanoflex system (Eksigent, AB SCIEX, Dublin, CA, USA city, state abbrev, USA) coupled to a Q-Exactive mass spectrometer (Thermo Fisher Scientific, San José, CA city, state abbrev, USA), through a 65 min gradient of 5–45% of eluent B (eluent A, 0.1% formic acid (Sigma-Aldrich Inc., St Louis, MO, USA company, city, state abbrev if USA, country) in water; eluent B, 0.1% formic acid in acetonitrile (Sigma-Aldrich Inc., St Louis, MO, USA company, city, state abbrev if USA, country)), at a flow-rate of 300 nL/min.

Full mass spectra were recorded in positive ion mode over a 400–1600 *m*/*z* range at a 70,000 FWHM (full width at half maximum) resolution, followed by 10 MS/MS spectra, at a resolution of 17,500 FWHM, generated in a data-dependent manner on the most abundant ions.

All data generated were searched using Proteome Discoverer 2.1 platform (Thermoscientific) based on SEQUEST search engine and human protein database (70,726 entries, downloaded on January 2017 from UNIPROT website, www.uniprot.gov) (accessed on 17 December 2020).

The obtained protein lists were aligned, normalized [30] and then processed by means of Linear Discriminant Analysis (LDA) [31]. For assigning each subject to a specific group, the α-value parameter was calculated according to the extracted marker proteins [32] (for more details, see Appendix A).

The cellular component enrichment of proteins in the examined conditions and the comparison versus the Vesiclepedia database were achieved using FunRich (version 3.1.3, company, city, state abbrev if USA, country) (open access standalone software downloadable from http://www.funrich.org/) (accessed on 17 December 2020).

### 4.6. Network Analysis

Starting from the list of proteins selected by differential and LDA procedures, the corresponding Homo sapiens Protein–Protein Interaction (PPI) network was extracted, and by means of the Cytoscape plug-in, STRING 8 database [33], known interactions were retrieved from several databases such as Prolink, DIP, KEGG and BIND. In addition, PPI was examined using Cytoscape 3.5 [34], and only experimentally verified interactions with >0.15 score were retained. Finally, Bingo 2.44 [16] was used to emphasize modules based on functionally organized gene ontology GO terms.

### 4.7. Statistical Analysis

The criteria for the identification of peptide sequences and related proteins were the following: trypsin was used as an enzyme; three missed cleavages permitted per peptide; mass tolerances of 10 ppm for precursor ions and ±0.05 Da for fragment ions were applied. A percolator node was used with target-decoy strategy to give final false discovery rates (FDR) at a Peptide Spectrum Match (PSM) level of 0.01 (strict) based on q-values, considering a maximum deltaCN of 0.05 [35]; a minimum peptide length of six amino acids and rank 1 were considered, and protein grouping and strict parsimony principle were applied.

Differences of categorical variables were evaluated using the X-squared test. The Kruskal–Wallis test was applied for comparing quantitative variables. Analyses were performed using R (3.5.2) statistical analysis software; a *p*-value < 0.05 was considered statistically significant.

A spectral counts (SpCs) based quantitative approach, although sometimes it is less accurate that other methods, is more easy to apply, and it is enough for statistical analysis if it is applied to evaluate differential expression comparing the same proteins and its value [36], combined with statistical analysis, such as by t-test, too [37]; proteins selected by both LDA and MAProMA (Multidimensional Algorithm Protein Map) were evaluated by hierarchical clustering [38,39], applying Ward’s method and the Euclidean distance metric. LDA using common covariance matrix for all stratified groups and Mahalanobis distance and hierarchical clustering were performed by JMP 5.1 software (SAS Institute, Cary, NC, USA company, city, state abbrev if USA, country)). To select proteins discriminating the stratified children groups, we considered those with the largest F ratio (≥5) and smallest *p*-value (≤0.01). Based on direct correlation between the spectral counts (SpC, also called PSM) and the relative abundance of identified proteins, DAve (Differential Average) and DCI (Differential Coefficient Index) indices of MAProMa software [4] were used to process the average (aSpC) corresponding to each analyzed children group. The threshold values imposed were DAve > |0.2| and DCI > |5|.

## 5. Conclusions

Protein fractionation, by the isolation of urinary extracellular vesicles, allowed an interesting statistically confident proteomic-based stratification of healthy preschool children. These samples are of primary importance to perform investigations involving children, reducing the invasiveness of collection; however, the biogenesis of urinary EVs remains unknown, so it will be important to investigate their biogenesis and release. The obtained childhood groups resulted in good agreement with the respiratory impedance and underlying biological profiles in the corresponding healthy preschool children. To our knowledge, the present study involves a higher number of healthy children investigated for urinary proteomics, and although these data are based on a limited number of subjects, they represent a proof-of-principle about the importance of an appropriate protein fractionation, such as extracellular vesicle purification from urine. Of course, future studies are needed to increase the number of analyzed subjects and to understand any roles of the selected proteins in the evaluation of physiological changes in lung function throughout childhood.

## Figures and Tables

**Figure 1 molecules-26-01258-f001:**
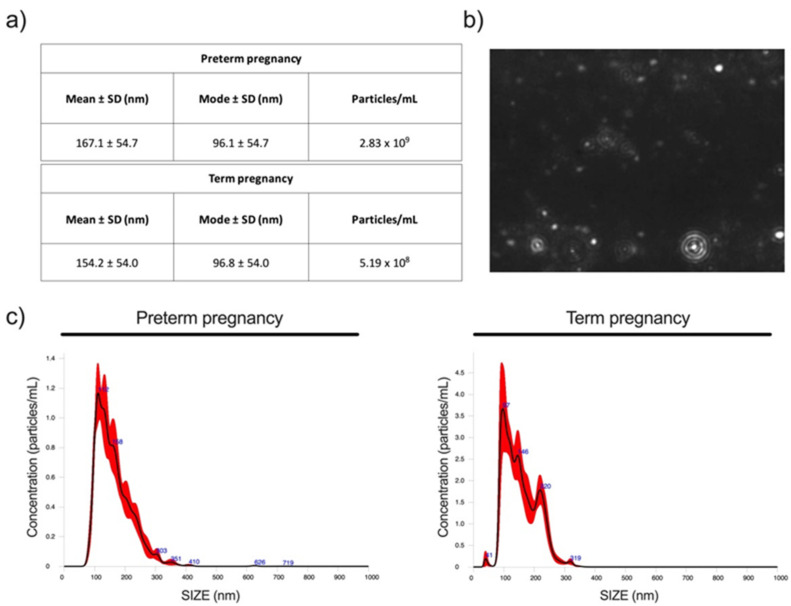
Schematic resume of the extracellular vesicle analysis carried out with nanosight NS300. The tables (**a**) collect the data regarding the measurements of preterm and term pregnancy; the highlighted data are the mean and the mode of the particles recorded by the instrument and the concentration per milliliter of the particles with a diameter lower than 200 nm. The picture (**b**) is a representative image taken during the extracellular vesicle analysis with nanosight. The graphs (**c**) are a simplified representation of the extracellular vesicle absolute size distribution and nanovesicle concentration per ml linked to their sizes; the red shadow is the standard deviation and the black line is the mean of 5 separate and consecutive analyses.

**Figure 2 molecules-26-01258-f002:**
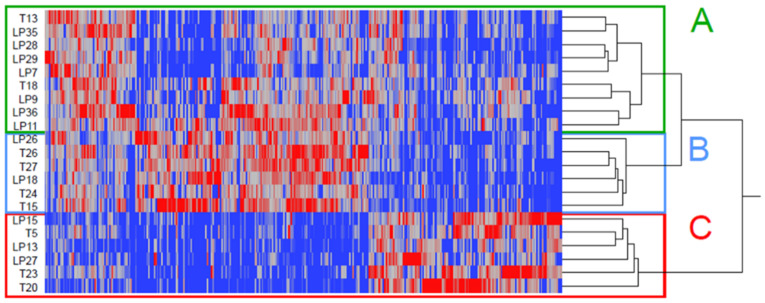
Linear Discriminant Analysis (LDA) placed children in three distinct groups called **A**, **B** and **C**.

**Figure 3 molecules-26-01258-f003:**
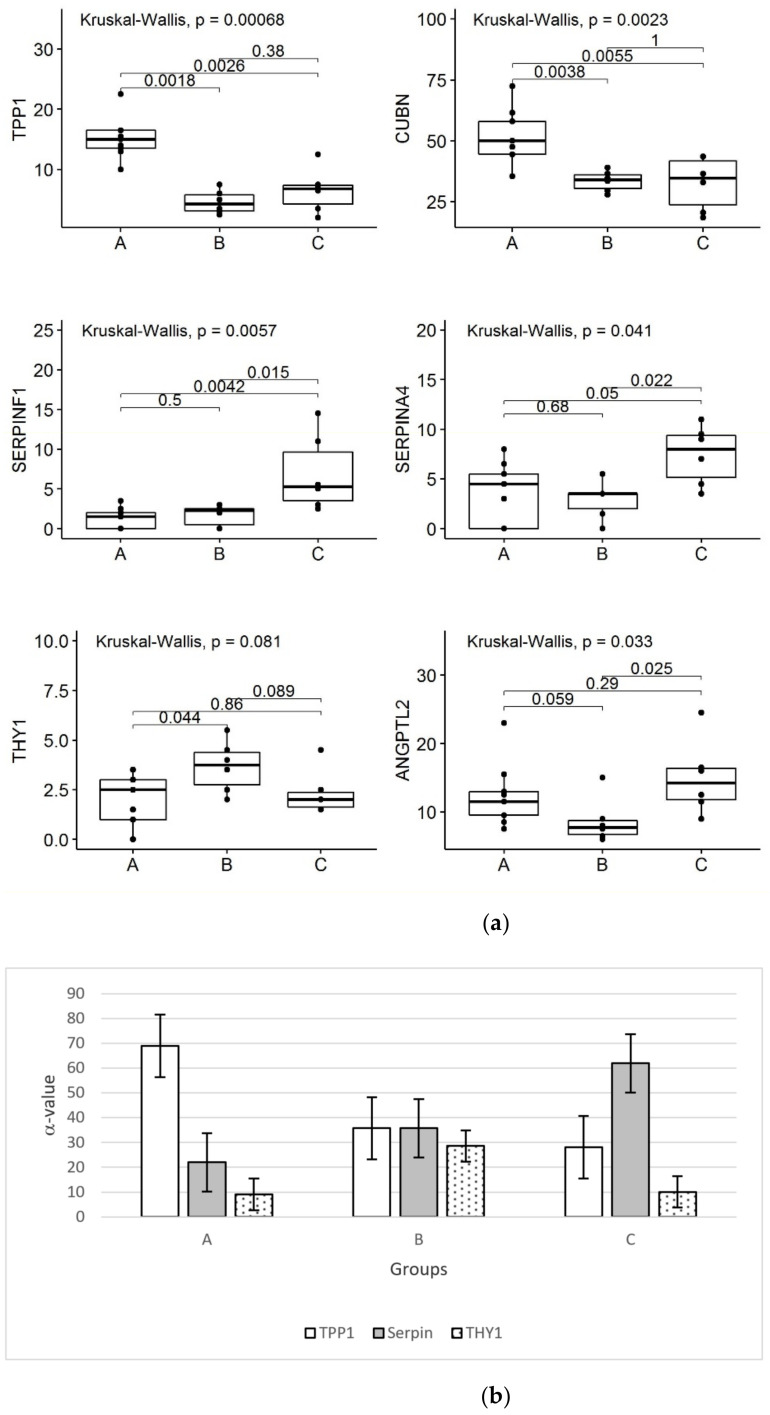
Relevant proteins useful to characterize the groups stratified by cluster analysis. (**a**) Boxplots of the 6 proteins (TPP1, CUBN, SERPINF1, SERPINA4, ANGPTL2 and THY1), expressed as spectral count, among groups A (*n* = 9 children), B (*n* = 6 children) and C (*n* = 6 children); (**b**) α-value calculation for three selected discriminant proteins.

**Figure 4 molecules-26-01258-f004:**
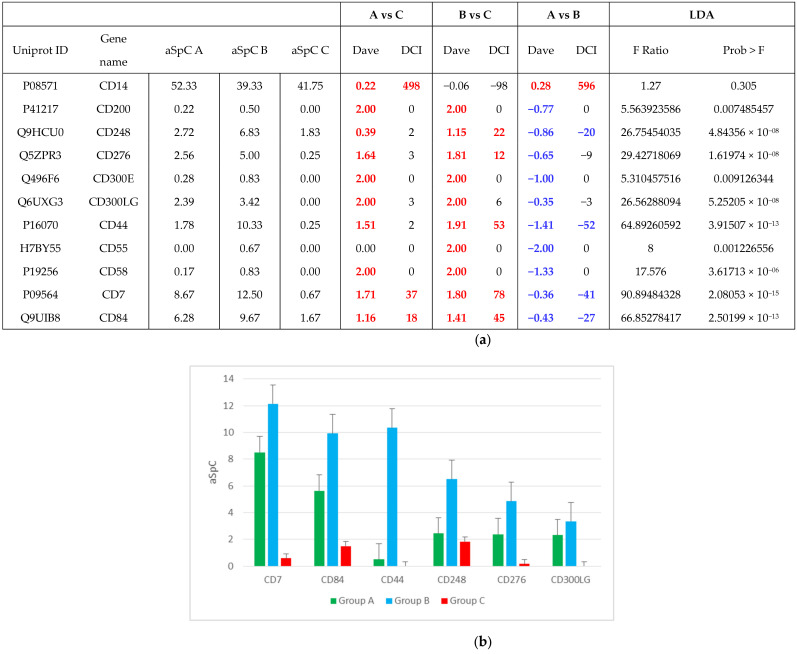
**(a**) Identified Cluster of Differentiation (CD), including average spectral count (aSpC), differential average (DAVE) and differential confidence index (DCI) values, and LDA; (**b**) Levels (aSpC) of more abundant CDs in the three groups (A, B and C). *Positive (red) and negative (blue) DAVE and DCI values indicate proteins upregulated in first and second item of comparison, respectively, representing confident values (higher than |0.2| and |10|, for DAVE and DCI, respectively).

**Figure 5 molecules-26-01258-f005:**
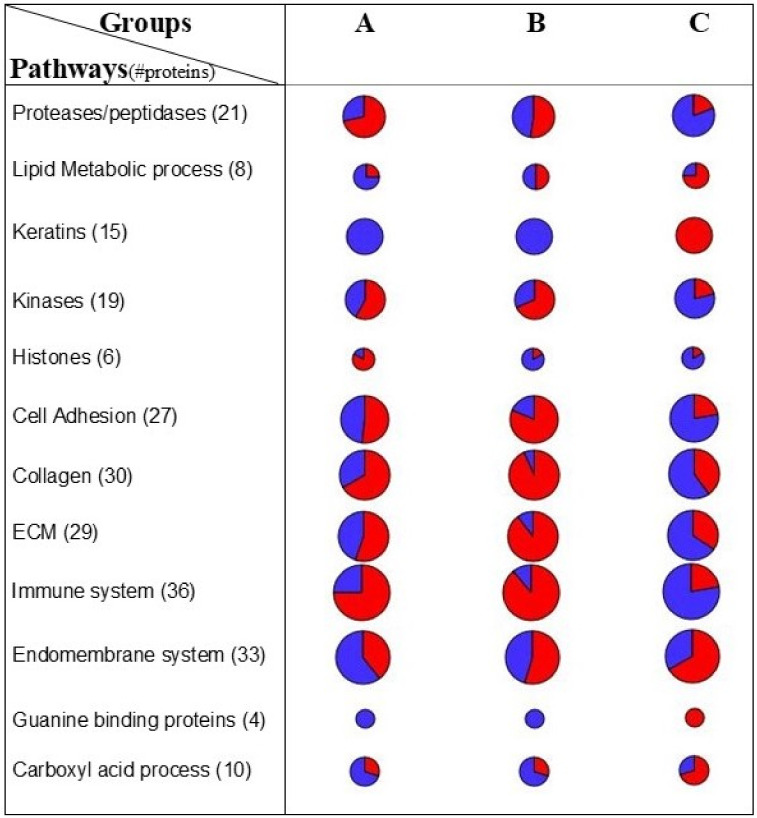
Protein expression per functional cluster. Enrichment of main functional categories of low (blue) and high (red) abundant proteins (Appendix A) in groups **A**, **B** and **C**. Bubble size is indicative of the number of proteins involved in each pathway. ECM stands for extracellular matrix.

**Figure 6 molecules-26-01258-f006:**
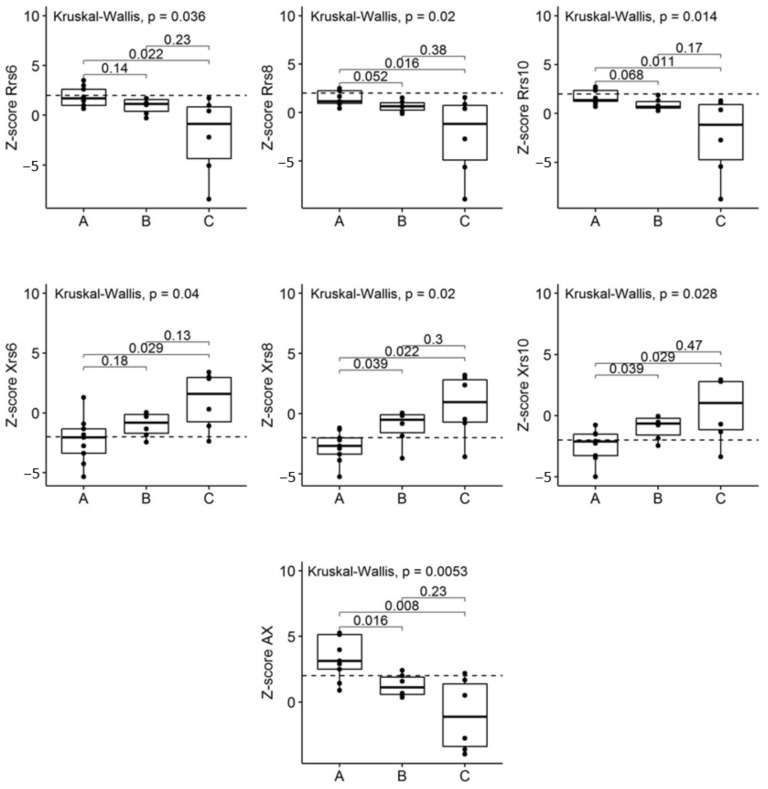
Baseline *Z-*scores of resistance at Rrs6, Rrs8, and Rrs10 Hz and reactance at Xrs6, Xrs8, and Xrs10 Hz and AX in the three groups A, B and C. *p*-values come from a Kruskal–Wallis test. On the vertical axes, the labels of FOT measures are reported, on the horizontal axes the labels of each group are reported. Dashed lines represent the normal limit.

**Table 1 molecules-26-01258-t001:** Subject characteristics between late preterm (LP) and term (T) in the study sub-sample (*n* = 21).

	LP	T	*p*-Value
	*n* = 12	*n* = 9	
*Personal characteristics*			
Female (%)	5 (41.67)	2 (22.22)	0.640
*Z*-score Height, mean (SD)	−0.92 (1.08)	0.21 (1.03)	0.033
*Z*-score BMI, mean (SD)	0.64 (1.47)	1.09 (1.09)	0.337
Age, years, mean (SD)	3.67 (0.78)	4.56 (0.73)	0.015
Birthweight, mean (SD), *gr*	2123.33 (576.78)	3012.22 (829.02)	0.012
Cesarean Delivery, *n* (%)	10 (83.33)	7 (77.78)	1.000
Maternal diseases in pregnancy, *n* (%)	2 (16.67)	0 (0.00)	0.592
Respiratory support, *n* (%)	5 (41.67)	3 (33.33)	1.000
Bronchiolitis 1st year of life, *n* (%)	4 (33.33)	4 (50.00)	0.780
Upper respiratory infection ever, *n* (%)	5 (41.67)	4 (44.44)	1.000
Pneumonia ever, *n* (%)	0 (0.00)	3 (33.33)	0.126
*Environmental exposure*			
Proximity to high traffic road <200 m, *n* (%),	10 (83.33)	9 (100.00)	0.592
Current parental smoke exposure, *n* (%),	4 (33.33)	0 (0.00)	0.173
Current pet exposure, *n* (%),	3 (25.00)	2 (22.22)	0.592
Current mold exposure, *n* (%),	2 (16.67)	0 (0.00)	0.592
*FOT parameters*			
*Z*-score			
Rrs6	0.98 (1.49)	−0.27 (3.79)	0.887
Rrs8	0.75 (1.34)	−0.71 (3.86)	0.619
Rrs10	0.92 (1.35)	−0.57 (3.85)	0.670
Xrs6	−1.36 (2.33)	−0.43 (2.23)	0.522
Xrs8	−1.64 (2.05)	−0.86 (2.54)	0.570
Xrs10	−1.55 (1.94)	−0.65 (2.22)	0.522
AX	2.06 (2.26)	0.80 (2.92)	0.356

Data are presented as *n* (%) or mean (SD). *X2 test was used for comparing frequencies; Kruskal–Wallis test for comparing quantitative variables, bold values are significant. FOT, Forced Oscillation Technique; Rrs6, respiratory system resistance at 6 Hz; Rrs8, respiratory system resistance at 8 Hz; Rrs10, respiratory system resistance at 10 Hz, Xrs6, respiratory system reactance at 6 Hz; Xrs8, respiratory system reactance at 8 Hz; Xrs10, respiratory system reactance at 10 Hz; Ax, area under the reactance curve.

## Data Availability

Proteomic data are available by ProteomeXchange Consortium repository at the site: ftp://massive.ucsd.edu/ MSV000086867/ (accessed on 17 December 2020).

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
