# Peer review of "Shotgun Proteomics of Isolated Urinary Extracellular Vesicles for Investigating Respiratory Impedance in Healthy Preschoolers"

_molecules, 2021, doi:10.3390/molecules26051258_

Round 1

Reviewer 1 Report

Authors examined the proteomes of exosomes isolated from urine using LC/MSMS shotgun method. Exosome proteomes were then correlated with patterns of respiratory impedance and possible protein biomarkers of different respiratory profiles were selected.

My concerns
Research design:
Exosomes released into the urine are derived from renal epithelial cells. Exosomes from other organs transferred by blood plasma are not likely to pass the glomerular filter due to their size. The size of isolated exosomes shown in Fig 1 is also greater than the pore size of the glomerular wall (12 nm, 10.1681/ASN.2010020199). Possible mechanisms of transfer of proteins in exosomes from airway cells to urine need to be discussed.

Methods:
Variability in urine composition occurs due to different hydration states, urinary flow, etc. In two groups of urine samples were detected differences in exosome dimension and count. Some normalization procedure (e.g urinary creatinine) should be applied to the proteomic data (10.1038/sj.ki.5000273).

Presentation of results:
Supplementary information (Table S1, S2, S3; Fig S1a,b, S2, S3, S4) are not included in the supplementary file.

Author Response

Reviewer  #1

Comments and Suggestions for Authors

Authors examined the proteomes of exosomes isolated from urine using LC/MSMS shotgun method. Exosome proteomes were then correlated with patterns of respiratory impedance and possible protein biomarkers of different respiratory profiles were selected.

Our Response - We thank the reviewer, for her/his appreciation and the interest in the topic of our paper,   concerning the proof of principle oriented to correlate clinical and molecular parameters.

Research design:

Exosomes released into the urine are derived from renal epithelial cells. Exosomes from other organs transferred by blood plasma are not likely to pass the glomerular filter due to their size. The size of isolated exosomes shown in Fig 1 is also greater than the pore size of the glomerular wall (12 nm, 10.1681/ASN.2010020199). Possible mechanisms of transfer of proteins in exosomes from airway cells to urine need to be discussed.

Our Response - the comment by reviewer is correct,  considering the usual application of EVs for investigating kindey-related diseases; however, recent studies evidenced that EVs can be used to investigate unrelated urogenital diseases, such as thyroid, bladder and pancreatic cancers  (Tse-Ying et al., Front Endocrinol  2020; Chen et al., J Proteome Res. 2012; Ferrari et al., Molecules 2019) asthma (Riccio et al, WAO J, 2020) and neurological diseases (Wang et al, EBioMedicine 2019), also. These studies detected typical markers of not-kindey diseases, such as thyroglobulin, gal-3, or  proteins linked to Parkinson's, Alzheimer's and Huntington's diseases.

As reported da Wang et al. (EBioMedicine 2019) “urinary EVs are an underutilized but highly accessible resource for biomarker discovery with particular promise”; this is of primary importance to perform investigations involving children, reducing the invasiveness of sample collection.

Of course, these findings involve that the glomerular filter based on size is not a sufficient mechanism to explain the secretion of not-kindey related urinary EVs. Important aspects of the biogenesis of Urinary EVs remain unknown, and in the next future it will be to explore about their biogenesis and release. Now we may only report registered that proteomic analysis of urinary EVs  was unexpected able to driven stratification of children, confirmed by clinical data.

Based on these comments, we have introduced new sentence in Discussion and Conclusion section, including references.

Methods:

Variability in urine composition occurs due to different hydration states, urinary flow, etc. In two groups of urine samples were detected differences in exosome dimension and count. Some normalization procedure (e.g urinary creatinine) should be applied to the proteomic data (10.1038/sj.ki.5000273).

Our Response - Creatinine is usually assayed to evaluate the urinary dysfunction. In our case, enrolled children were not affected by kindey problems and for this reason it was not scheduled to measure creatinine level. However, we performed normalization at two levels: firstly, by injection of fixed total amount of proteins (in our case 1 microgram) and secondly, by means of post-acquisition procedure of SpCs, according the procedure reported in the Method section (details reported in reference #31, and based on the total spectral count per sample has to be equal (so called “total signal normalization”). See  our response to Reviewer #2, also.

Presentation of results:

Supplementary information (Table S1, S2, S3; Fig S1a,b, S2, S3, S4) are not included in the supplementary file.

Our Response - We apologize for the missed supplementary information; effectively there is a red X on the supplementary section of Journal website, although the zip file may be downloaded. In the revised version we check the correct upload. To avoid problems, we send the supplementary files as attachment to the editor, also.

Reviewer 2 Report

In this manuscript, the authors identified several proteomic signatures of respiratory impedance in healthy preschoolers from urine extracellular vesicles. While the aim of this study is interesting and the manuscript is well written, the manuscript raised several concerns. The authors should take into account the concerns listed below before considering the manuscript for publication in MOLECULES.

GENERAL COMMENTS

  • There is no proof that the authors isolated exclusively exosomes and no other extracellular vesicles. So, the authors should talk about extracellular vesicles rather than exosome. Moreover, as mentioned below the authors should check if the quantified proteins are known to be in the extracellular vesicles by checking it with protein databases of extracellular vesicles (such as vesiclepedia) and performing a gene ontology analysis on the protein known localization.
  • At the beginning of the manuscript the authors mentioned to work on a cohort of 66 patients while the manuscript deals with a proteomic study done on only 21 patients. The authors should not talk about the 66 patients as the proteomic data are obtained for 21 patients of those 66 patients and no proteomic data were described on the other patients. I still agree with the conclusions even if the number of patient is lower.
  • The DEP data are obtained only from label-free spectral count proteomic approach. There is no complementary analysis to confirm the DEP of a set of proteins. The authors should present the complementary data or mentioned in the manuscript the limit of the study.

ABSTRACT

No comment

INTRODUCTION

No comment

RESULTS

  • Lines 94-95 “In the total sample (n=66), no differences were found for Z-scores of Rrs and Xrs at 6, 8, and 10 Hz and AX, between late preterm (LP) and term (T) either (Data not shown).” Then, the authors performed the proteomic study on 21 patients by claiming that they are representative to the 66. The table S1 seems to confirm it that there is no difference between the 21 patient, the 66 patients and the 45 patients. The authors should indicate it. Nevertheless, I’m not comfortable with method. It would have been more accurate to indicate directly that this analysis was done on 21 patients and then discuss about the extrapolation on the 66 patients.
  • Figure 5: for more clarity, the authors should indicate the number of proteins on the red and blue bubles
  • To confirm that urine preparation allowed the authors to investigate the proteins from extracellular vesicles of urines, the authors should i) compare the list of the 1127 proteins they identified/quantified to the ones reported in the databases of extracellular vesicles such as vesiclepedia and ii) perform a gene ontology analysis to check the known localization of proteins.
  • Figure 6: what are the differences between each box plot figure?

SUPPLEMENTARY MATERIALS

  • The authors should indicate what is the volume of the urine collected and what is the volume of urine used for the proteomic study. They should also indicate if they extracellular vesicles are from the morning urine, second urine or middle-day urine
  • In proteomic data processing, the authors applied a mass tolerances of 10 ppm for precursor ions and ±0.6 Da for fragment ions. A mass tolerance 0.6 Da seems very large while the resolution was the resolution was 17,500 FWHM.
  • In tables S2 and S3, the number of unique peptides used for quantitation should be indicated. Moreover, the quantitation is usually considered for proteins with at least 2 unique peptides. The authors didn’t mention anything about it. They should take it into account.
  • Line 12-125: Why the authors choose these filtering parameters (F ratio>5 and p-value < 0.01) for the significant differentially expressed proteins? It would be appreciating if the authors present a volcano plot at least in supplemental data of the –ln p value vs log fold change to determine if the threshold mention above are appropriate.
  • In the sentence “Specifically, the threshold values imposed were DAve >|0.2| and DCI>|100|” the authors should precise for what it was imposed to refine de DEP?

Author Response

Reviewer #2

Comments and Suggestions for Authors

In this manuscript, the authors identified several proteomic signatures of respiratory impedance in healthy preschoolers from urine extracellular vesicles. While the aim of this study is interesting and the manuscript is well written, the manuscript raised several concerns. The authors should take into account the concerns listed below before considering the manuscript for publication in MOLECULES.

GENERAL COMMENTS

There is no proof that the authors isolated exclusively exosomes and no other extracellular vesicles. So, the authors should talk about extracellular vesicles rather than exosome. Moreover, as mentioned below the authors should check if the quantified proteins are known to be in the extracellular vesicles by checking it with protein databases of extracellular vesicles (such as vesiclepedia) and performing a gene ontology analysis on the protein known localization.

At the beginning of the manuscript the authors mentioned to work on a cohort of 66 patients while the manuscript deals with a proteomic study done on only 21 patients. The authors should not talk about the 66 patients as the proteomic data are obtained for 21 patients of those 66 patients and no proteomic data were described on the other patients. I still agree with the conclusions even if the number of patient is lower.

The DEP data are obtained only from label-free spectral count proteomic approach. There is no complementary analysis to confirm the DEP of a set of proteins. The authors should present the complementary data or mentioned in the manuscript the limit of the study.

Our Response - We thank the reviewer for her/his general and specific comments; below we reply point to point. As suggested we changed exosome with extracellular vesicles (EVs); in fact, isolation by ultracentrifugation allows a wide range of vesicles. Concerning the limit of the study, we already reported in the first version of the manuscript that the lack of external validation and the small sample size are the main limitations of the current study. However, the present work represent a preliminary proof of principle based on integration of respiratory functions and urinary proteomics on healthy pre-schoolers.

RESULTS

Reviewer comment - Lines 94-95 “In the total sample (n=66), no differences were found for Z-scores of Rrs and Xrs at 6, 8, and 10 Hz and AX, between late preterm (LP) and term (T) either (Data not shown).” Then, the authors performed the proteomic study on 21 patients by claiming that they are representative to the 66. The table S1 seems to confirm it that there is no difference between the 21 patient, the 66 patients and the 45 patients. The authors should indicate it. Nevertheless, I’m not comfortable with method. It would have been more accurate to indicate directly that this analysis was done on 21 patients and then discuss about the extrapolation on the 66 patients.

Our Response -  We understand the Reviewer’s concern and have now reworded and implemented the corresponding sentences in the Results section; as required we have underlined it and directly indicated that proteomics analysis concerns 21 subjects out of 66.

Reviewer comment - Figure 5: for more clarity, the authors should indicate the number of proteins on the red and blue bubbles

Our Response - The requirement of reviewer is right; in the new Figure 5 we added the number of proteins for each pathway to improve readability of Figure 5; while red and blue parts are a percentage of entire, represented by the total number of proteins for each category.

Reviewer comment - To confirm that urine preparation allowed the authors to investigate the proteins from extracellular vesicles of urines, the authors should i) compare the list of the 1127 proteins they identified/quantified to the ones reported in the databases of extracellular vesicles such as vesiclepedia and ii) perform a gene ontology analysis to check the known localization of proteins.

Our Response - As required, using FunRich tool, we compared the protein lists for obtained from each stratified group in versus Vesiclepedia database (http://microvesicles.org/),. Moreover we performed including the gene ontology analysis to check the localization of proteins, also. All the obtained results are reported in the See new Supplementary Figure S3.

Reviewer comment - Figure 6: what are the differences between each box plot figure?

Our Response - We thank the Reviewer for the question. Each box plot reports the value for the seven considered FOT parameters, referred to resistance and reactance. We added more details in the caption of the figure and in the main text (2.3 Linking FOT parameters and proteomics section).

SUPPLEMENTARY MATERIALS

Reviewer comment - The authors should indicate what is the volume of the urine collected  and what is the volume of urine used for the proteomic study. They should also indicate if they extracellular vesicles are from the morning urine, second urine or middle-day urine.

Our Response - We apologize the missed information. Now, according the Reviwer’ suggestion, in the supplementary material (see 4.3 Isolation of urinary EVs section), we specified the volume (100 ml) of the  morning urine processed and the aliquot (5 mL) used for EV isolation.

Reviewer comment - In proteomic data processing, the authors applied a mass tolerances of 10 ppm for precursor ions and ±0.6 Da for fragment ions. A mass tolerance 0.6 Da seems very large while the resolution was the resolution was 17,500 FWHM.

Our Response – We apologize our typing error; the correct mass tolerance set for fragment ions was ±0.05 Da, as shown in the processing workflow screenshot reported below.

Reviewer comment - In tables S2 and S3, the number of unique peptides used for quantitation should be indicated. Moreover, the quantitation is usually considered for proteins with at least 2 unique peptides. The authors didn’t mention anything about it. They should take it into account.

Line 12-125: Why the authors choose these filtering parameters (F ratio>5 and p-value < 0.01) for the significant differentially expressed proteins? It would be appreciating if the authors present a volcano plot at least in supplemental data of the –ln p value vs log fold change to determine if the threshold mention above are appropriate.

In the sentence “ Specifically, the threshold values imposed were DAve >|0.2| and DCI > |100|” the authors should precise for what it was imposed to refine the DEP?

Our Response - As required by Reviewer, we have added the number of unique peptides in  tables S2 and S3.

We would clarify that. LC-MS raw data were processed by a licensed and commercial software, Proteome Discoverer v. 2.1, and the identified proteins were validated using the Percolator tool, that assigns reliable statistical confidence measures. Percolator is a superior validation algorithm which exploits q-value that increases the identification confidence integrating p-value with statistical estimation of true posterior error probabilities based on a target-decoy strategy to give, as set by us, a final false discovery rates at peptide spectrum match level of 0.01.

Moreover, we added further filters to increase the identification confidence: a stringent mass tolerance both on precursor and fragment ions, protein grouping and strict parsimony principle (all proteins that have not been identified with at least one unique peptides are removed), a minimum peptide length of six amino acids, and rank 1 (i.e. only peptides placed in first position database search were extracted to avoid the redundant assignment of a spectrum to different peptides/proteins). Applying all the imposed thresholds, for quantitative analysis we considered proteins with at least one unique peptide but identified at least in two runs.

More important, our quantitative approach is based on spectral count evaluation and not peak area; since we compared the three different groups of subjects, we maintained also proteins identified at low level (SpC <2). The reason of this procedure is due to the application of linear discriminant analysis which allows the reduction of data metrics dimensionality and the extraction of proteins that best discriminate the investigated groups of subjects. The threshold applied to select those proteins (called descriptors of groups) was exclusively set to p-value < 0.01, corresponding to Fisher ratio (F) > 5 (according to JMP 15.2 Sas software).

In addition, we further filtered the most significant differentially expressed proteins by applying DAve and DCI algorithms, included in the home-made MAProMa software (De Palma et al., mBio 2020; Riccio et al., WAO Journal 2020; Sereni et al., JACI 2018; Mauri et al. Methods Enzymol. 2008), on the aSpC of proteins belonging to the three examined groups, already statistically filtered and averaged, in order to extract high stringent differentially expressed proteins.; In particular DAve allows a value comparable to ln[FC]; of note FC requires an arbitrary value for protein exclusively identified in one sample, to avoid non-sense values (such as n/0 or 0/n); on the contrary, DAve returns always a finite value. The high overlap of proteins extracted by the two independent methodologies, reported in Table S3, and the 15 proteins, highlighted in the manuscript and on which the quantitative speculation is focused, proved, both in terms of statistical and differential analysis, the validity of our approach.

Finally, LDA and MAProMa thresholds are fixed in advance, independently from the experimental values, to avoid subjective evaluations. For these reasons, it is not informative to make a volcano plot because the proteins reported in Table S3 are already statistically filtered. However, to meet Reviewer’ request, we prepared a volcano-like diagram (new Supplementary Figure S4), plotting DAve vs DCI values of proteins extracted by linear discriminant analysis (p-value < 0.01), in order to highlight the range of proteins shared between LDA and DEP analyses. Based on our experience DAve and DCI thresholds of 0.2 and 5, respectively, combined to LDA statistical filtering allow a good stringency, when normalized values are used.

Reviewer 3 Report

The manuscript of Ferrante et al. with the title “Shotgun proteomics of isolated urinary exosomes for investigating respiratory impedance in healthy preschoolers” aims using proteomics for urinary exosome analysis to characterize different patterns of respiratory impedance in healthy preschoolers. The authors use an experimental approach using exosome isolation by ultracentrifugation, state-of-the-art MS and appropriate statistical and bioinformatics tools for data processing.

Major concerns

Based on proteomics obtained data, authors stratified 3 groups of children (A, B, C), with significantly different patterns of respiratory impedance; however, they should clarify whether any relationship was found between those groups found and the subpopulations used in the study late preterm (LP) and full-term (T) born.

As recognized by the authors “The lack of external validation and the small sample size are the main limitations of the current study”.

Minor concerns

69   spirometry spirometry – repeated word

79 the term microvesicles is not synonymous with exosomes

83 heal-thy  substitute by healthy

145 Sepin or Serpin?

Author Response

Reply to Reviewer #3

Comments and Suggestions for Authors

The manuscript of Ferrante et al. with the title “Shotgun proteomics of isolated urinary exosomes for investigating respiratory impedance in healthy preschoolers” aims using proteomics for urinary exosome analysis to characterize different patterns of respiratory impedance in healthy preschoolers. The authors use an experimental approach using exosome isolation by ultracentrifugation, state-of-the-art MS and appropriate statistical and bioinformatics tools for data processing.

Major concerns

Reviewer comment - Based on proteomics obtained data, authors stratified 3 groups of children (A, B, C), with significantly different patterns of respiratory impedance; however, they should clarify whether any relationship was found between those groups found and the subpopulations used in the study late preterm (LP) and full-term (T) born. As recognized by the authors “The lack of external validation and the small sample size are the main limitations of the current study”.

Our Response - We thank the Reviewer for the comment. In the discussion section, we have added the following sentence “Here we report the application of an innovative approach such as urinary proteomics to characterize the respiratory impedance in healthy preschoolers. Using urine proteomic analysis, we linked to underlying biological profiles different patterns of respiratory impedance observed in healthy preschoolers. Of note, the three proteome-based groups had significantly different resistance, reactance and AX values measured by FOT, regardless of being LP or T.”

Minor concerns

69 spirometry spirometry – repeated word

79 the term microvesicles is not synonymous with exosomes

83 heal-thy substitute by healthy

145 Sepin or Serpin?

Our Response - We apologize for these typing errors; we thank Reviewer for underlining them; in the revision version these issues were fixed.

Round 2

Reviewer 1 Report

Authors answered comments and included included their response to the manuscript file. I suggest to accept the manuscript.

Reviewer 2 Report

The authors properly edited their manuscript according to the concerns I raised in my previous review.

In my point of view, that this revised manuscript may be considered in the current form for publication in MOLECULES.